# DynaPrompt: Dynamic Test-Time Prompt Tuning

**Zehao Xiao[1], Shilin Yan[2], Jack Hong[2], Jiayin Cai[2], Xiaolong Jiang[2], Yao Hu[2],**
**Jiayi Shen[1],[*] Qi Wang[3], Cees G. M. Snoek[1]**
[1]AIM Lab, University of Amsterdam
[2]Xiaohongshu Inc. [3]Department of Automation, Tsinghua University

## Abstract

Test-time prompt tuning enhances zero-shot generalization of vision-language models but tends to ignore the relatedness among test samples during inference. Online test-time prompt tuning provides a simple way to leverage the information in previous test samples, albeit with the risk of prompt collapse due to error accumulation. To enhance test-time prompt tuning, we propose DynaPrompt, short for *dynamic test-time prompt tuning*, exploiting relevant data distribution information while reducing error accumulation. Built on an online prompt buffer, DynaPrompt adaptively selects and optimizes the relevant prompts for each test sample during tuning. Specifically, we introduce a dynamic prompt selection strategy based on two metrics: prediction entropy and probability difference. For unseen test data information, we develop dynamic prompt appending, which allows the buffer to append new prompts and delete the inactive ones. By doing so, the prompts are optimized to exploit beneficial information on specific test data, while alleviating error accumulation. Experiments on fourteen datasets demonstrate the effectiveness of dynamic test-time prompt tuning.[1]

## 1 Introduction

Despite achieving remarkable successes, foundation models such as Contrastive Language-Image Pretraining (CLIP) (Radford et al., 2021) still suffer from distribution shifts when adapting to downstream tasks (Zhou et al., 2022a;b; Xiao et al., 2024). To improve test-time adaptation of the model in the presence of distribution shifts, recent works introduce learnable prompts at test time. The methods freeze the CLIP model parameters while only tuning the learnable prompts for test data. As shown in Figure 1a, test-time prompt tuning (TPT) (Shu et al., 2022) adapts the prompt to each test sample individually, which is widely followed by recent works (Ma et al., 2023; Samadh et al., 2023; Yoon et al., 2024). However, tuning in such a way ignores the relatedness among test samples, which offers rich information on the test data distribution. To incorporate the information from relevant test samples, one straightforward method is to follow previous test-time adaptation methods (Wang et al., 2021; Goyal et al., 2022) and tune the test prompts online (Figure 1b). This encourages the prompts to exploit previous test information for better model adaptation. However, as detailed later on in this paper, we establish that online test-time tuning leads to severe prompt collapse due to error accumulation.

To have test-time prompt tuning benefit from relevant online information while reducing error accumulation, we constitute the concept of *dynamic test-time prompt tuning*, abbreviated as DynaPrompt. Specifically, DynaPrompt adaptively selects and optimizes the relevant online prompts for each sample while freezing the rest, yielding effective adaptation for the entire test set. As illustrated in Figure 1c, a prompt buffer is involved, which enables a set of online prompts to be flexibly selected, updated, and appended for each test sample. To ensure the appropriate prompt selection without collapse in DynaPrompt, we devise a comprehensive selection strategy with two metrics: *prediction entropy* and *probability difference*, which measure the model uncertainty of the predictions and the model sensitivity to the input changes. Then we construct the prompt screening threshold from these two metrics to achieve adaptive selection for different test samples. Such a screening rule selects the prompts with lower prediction entropy and larger probability differences and prefers those with

---

[*]Corresponding author.
[1]Codes are available at `https://github.com/zzzx1224/DynaPrompt`.

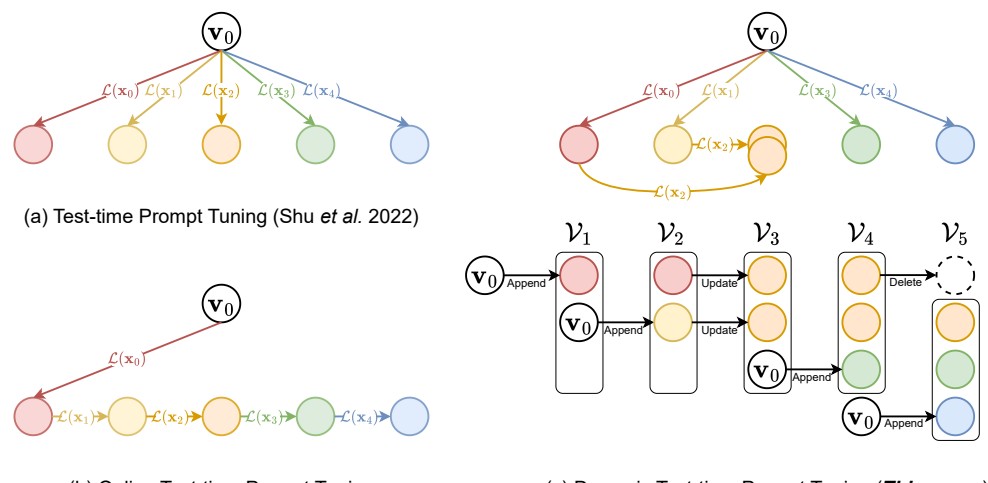

Figure 1: **Illustrations of different test-time prompt tuning methods.** Circles with specific colors denote learned prompts for individual test samples. For a single test sample, (a) test-time prompt tuning learns prompts from a shared initialization $\mathbf{v}_0$, ignoring the relatedness among test samples. (b) online test-time prompt tuning incorporates previous test sample information by using the previous-sample optimized prompt as the starting point, which leads to error accumulation. (c) we propose dynamic test-time prompt tuning (top) to adaptively exploit relevant information from previous test samples and alleviate error accumulation, which is achieved by autonomously selecting, updating, appending, and deleting online prompts in a prompt buffer $\mathcal{V}$ (bottom).

higher certainty in the sample predictions and more sensitivity to structural changes of the input. As a result, these selected prompts tend to be more relevant to the test sample while preventing collapse.

To incorporate new prompts in the prompt buffer for unexplored test data information, we include the operation of dynamic prompt appending in DynaPrompt. In the case of no relevant prompts available in the prompt buffer, DynaPrompt autonomously appends new online prompts and deletes the inactive ones. Through adaptively selecting, updating, appending, and deleting prompts, the online prompts are tailored and optimized for test samples with relevant data information. In this way, the predictions of the test samples are enhanced by leveraging relevant online information, while the error accumulation is reduced by flexible prompt updates in the prompt buffer.

Empirically, we conduct experiments on fourteen benchmarks, covering typical evaluation scenarios such as domain generalization and cross-dataset. The results show the effectiveness of the proposed method. Moreover, our method can be seamlessly incorporated into most prompt-tuning methods (Zhou et al., 2022b; Khattak et al., 2023) to enhance their performance further.

## 2 PRELIMINARY

We first provide a brief preliminary on the CLIP model, prompt learning, and test-time prompt tuning, containing background and commonly used techniques for test-time prompt tuning.

**CLIP model** (Radford et al., 2021). This pretrained model consists of an image encoder $\mathcal{F}_{\boldsymbol{\theta}_I}(\cdot)$ and a text encoder $\mathcal{F}_{\boldsymbol{\theta}_T}(\cdot)$, where $\boldsymbol{\theta}_I$ and $\boldsymbol{\theta}_T$ denote pre-trained parameters of the corresponding encoders. The image and text encoders take an input image $\mathbf{x}$ and text prompts as inputs, respectively. Given a downstream classification task with a set of class names, CLIP performs zero-shot classification on each input image $\mathbf{x}$. Specifically, CLIP gets the image feature as $\mathbf{f}_\mathbf{x} = \mathcal{F}_{\boldsymbol{\theta}_I}(\mathbf{x})$ and text features (i.e., zero-shot classifier) as $\{\mathbf{f}_{\mathbf{t}_c} | \mathbf{f}_{\mathbf{t}_c} = \mathcal{F}_{\boldsymbol{\theta}_T}(\mathbf{t}_c)\}_{c=1}^C$. $C$ is the number of class names and $\mathbf{t}_c$ is a manual-crafted text prompt corresponding to class $c$, e.g., "*a photo of a [class c].*" The probability of $\mathbf{x}$ belonging to class $c$ is $p(\hat{y} = c|\mathbf{x}) = \frac{\exp(\cos(\mathbf{f}_\mathbf{x}, \mathbf{f}_{\mathbf{t}_c})/\tau)}{\sum_{c=1}^C \exp(\cos(\mathbf{f}_\mathbf{x}, \mathbf{f}_{\mathbf{t}_c})/\tau)}$, where $\cos(\cdot, \cdot)$ denotes cosine similarity and $\tau$ is a learned temperature. Thus, CLIP directly obtains the prediction as $\arg\max_c p(\hat{y} = c|\mathbf{x})$.

**Prompt learning.** To further enhance the adaptation of CLIP on downstream tasks, recent methods, e.g. (Zhou et al., 2022a;b; Khattak et al., 2023), introduce learnable prompts $\mathbf{v} = [v_1][v_2]\cdots[v_n]$, while freezing the parameters of CLIP's encoders. Zhou et al. (2022b) introduce learnable prompts

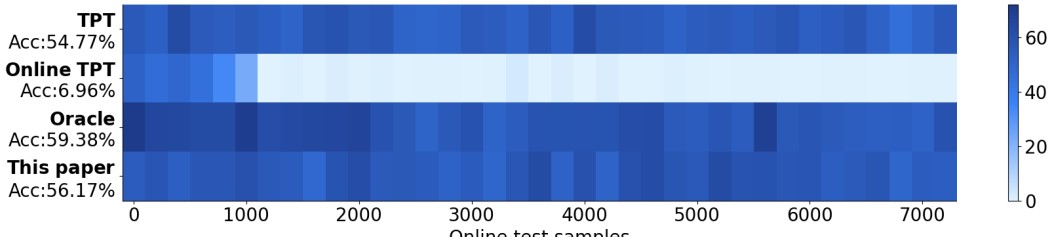

Figure 2: **Prompt collapse in online test-time prompt tuning.** We measure online test-time accuracy for different methods on ImageNet-A, where the accuracy for each block is calculated on 200 samples. Test-time Prompt Tuning (TPT) achieves stable accuracy by independently tuning the prompts for each test sample. Online TPT has severe error accumulation problems with competitive performance at the beginning while dropping significantly during online learning. Oracle is more stable by online tuning prompts only for correct predictions and achieves much better performance by incorporating the relevant information. Our method aims to exploit relevant information from previous online samples automatically while reducing error accumulation.

$\mathbf{v}$ to change the "*a photo of a*", making the prompts as $\mathbf{t}_c = [\mathbf{v}][class\ c]$. For prompt tuning, these methods rely on a small-scale training set of the downstream task containing input images and their corresponding class names, i.e., $\{\mathbf{x}, y_{gt}\}$, where $y_{gt} \in \{1, 2, ..., C\}$ is the ground-truth. During training, the model optimizes the learnable prompts $\mathbf{v}$ by minimizing the cross-entropy loss, i.e., $\min_{\mathbf{v}} -\log p(\hat{y} = y_{gt}|\mathbf{x}, \mathbf{v})$.

**Test-time prompt tuning.** Due to the existence of distribution shifts at test time, conventional prompt tuning methods usually suffer from overfitting, thus degrading the generalization ability of CLIP (Xiao et al., 2024). To address this issue, test-time prompt tuning (Shu et al., 2022; Samadh et al., 2023; Ma et al., 2023) are developed to independently tune each test sample's learnable prompts at test time. Here, every sample tunes the prompts from a shared initial state $\mathbf{v}_0$, which can be manually crafted (Shu et al., 2022) or pre-trained (Zhou et al., 2022b; Khattak et al., 2023). Since the labels are not available at test time, the optimization of prompts is guided by the prediction entropy of the selected sample augmentations:

$$\min_{\mathbf{v}} \mathcal{L}_{ent}(\mathbf{v}; \mathbf{x}_n) = \min_{\mathbf{v}} -\sum_{c=1}^{C} p(\hat{y} = c|\mathbf{X}_n, \mathbf{v}) \log p(\hat{y} = c|\mathbf{X}_n, \mathbf{v}), \tag{1}$$

where $\mathbf{X}_n$ denotes selected augmentations with lower prediction entropy for a test sample $\mathbf{x}_n$ (Shu et al., 2022; Samadh et al., 2023) and $p(\hat{y} = c|\mathbf{X}_n, \mathbf{v})$ is an averaged prediction probability across selected augmentations.

It is worth noting that most test-time tuning methods independently adjust the prompt for each test sample, ignoring the exploitation of relevant information in previous samples. To explore the benefits of the relevant information, we dive into test-time prompt tuning in an online manner.

## 3 PROMPT COLLAPSE IN ONLINE PROMPT TUNING

In several real-world applications, there are a large number of related test samples that arrive sequentially. In such cases, earlier observed samples can provide beneficial information about the test distribution and reserve the potential to improve the prediction for subsequent samples. Inspired by this insight, we propose to extend test-time prompt tuning (Shu et al., 2022) to online scenarios, formulating online test-time prompt tuning (*Online TPT*).

Online TPT retains most of the setups in TPT (Shu et al., 2022), where prompts are obtained through one-step optimization using entropy minimization for each test sample. As illustrated in Figure 1, the primary distinction lies in the initialization of the prompt for each test sample. While TPT (Shu et al., 2022) resets the prompt to the initial state $\mathbf{v}_0$ for each sample, Online TPT uses the optimized prompt from the previous sample as the starting point for the current sample.

As shown in Figure 2, Online TPT performs competitively with TPT initially, but the performance declines rapidly, nearly approaching 0% at the end. This work refers to this phenomenon as *prompt collapse*, where the prompt tends to accumulate excessive noise and prevent the model from making

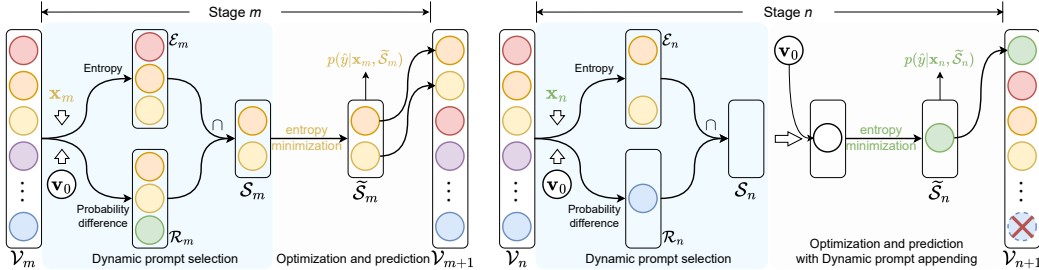

Figure 3: **The process of our dynamic test-time prompt tuning.** (Left) In dynamic prompt selection, we select the relevant online prompts from the prompt buffer $\mathcal{V}$ for each test sample using the intersection of the prompt subsets obtained by *entropy* and *probability difference* metrics. The selected prompts are optimized by entropy minimization before making predictions. (Right) If no prompt is selected, our dynamic prompt appending strategy assigns a new prompt initialized by $\mathbf{v}_0$ for the test sample and appends it to the prompt buffer. We always append the new optimized prompt on top of the buffer, moving the inactive ones to the bottom, which we can remove directly when appending new prompts to the full buffer.

accurate predictions. We attribute this to the entropy-based objectives, which might drive the optimization in the wrong directions without ground truth labels (Lee et al., 2024). The errors introduced during these optimization steps accumulate throughout online learning. Such *error accumulation* causes the prompts to progressively degenerate, resulting in gradually degraded performance in Figure 2. Notably, the existence of *error accumulation* will ultimately cause collapsed prompts that generate incorrect predictions with lower entropies.

To further understand the relevant online information's influence on prompt collapse, we implement an *Oracle* method for online test-time prompt tuning. The method follows most setups in Online TPT. By introducing the ground truth, Oracle only updates the prompts with correct predictions and skips the incorrect ones, thereby circumventing error accumulation from incorrect predictions. Still, entropy minimization works as the objective function for prompt tuning, the same as TPT and Online TPT. In Figure 2, Oracle considerably outperforms online TPT, confirming that error accumulation of entropy minimization with incorrect predictions hurts online test-time prompt tuning. Meanwhile, Oracle also outperforms TPT, which implies that the relevant information in online test samples benefits prompt tuning on the test distribution. All of the above observations motivate us to develop DynaPrompt in this work.

## 4 DYNAMIC PROMPT TUNING

As previously mentioned, this work develops DynaPrompt to enrich the family of test-time prompt tuning. Our motivation is to exploit beneficial information from prompt histories and alleviate the error accumulation in online prompt tuning. DynaPrompt adaptively selects relevant online prompts for each test sample to optimize and includes an online update prompt buffer $\mathcal{V}_n$ for each specific test sample $\mathbf{x}_n$. The buffer contains a set of online learnable prompts $\mathcal{V}_n = \{\mathbf{v}_i\}_{i=1}^{M_n}$ to store the distribution information in past samples, where $M_n$ denotes the number of prompts in the buffer at test step $n$. Each online prompt $\mathbf{v}_i$ is initialized with a hand-crafted prompt (Shu et al., 2022) or a pretrained prompt (Zhou et al., 2022b) $\mathbf{v}_0$ and adaptively optimized with the online samples. During prompt tuning, each test sample first selects a subset of the online prompts in the buffer and updates the buffer by optimizing the selected prompts. As shown in Figure 3, DynaPrompt consists of dynamic prompt selection (Section 4.1) and prompt appending (Section 4.2) strategies, as well as optimization and prediction with the dynamic prompts.

### 4.1 DYNAMIC PROMPT SELECTION

DynaPrompt introduces the dynamic prompt selection strategy to select appropriate prompts for each test sample from the online prompt buffer. Such a strategy returns a subset of the selected prompts $\mathcal{S}_n = \{\mathbf{v}_i \in \mathcal{V}_n \mid f(\mathbf{v}_i)\}$ for sample $\mathbf{x}_n$, where $f(\mathbf{v}_i)$ denotes selection conditions with specific metrics. We introduce the prediction entropy and probability difference as the selection metrics.

**Entropy-based selection.** We employ the prediction entropy as one of our prompt selection metrics. Widely used in classification tasks, entropy quantifies the uncertainty of predictions, assessing how confident the prompt is on the test data. Prompts with lower entropies reflect more confident predictions on the test sample (Niu et al., 2022; 2023), indicating the prompt has more prior knowledge and relevant information on the sample. Given a test sample $\mathbf{x}_n$ and the online prompt in the corresponding prompt buffer $\mathbf{v}_i \in \mathcal{V}_n$, we can calculate the entropy as:

$$\mathcal{D}_{ent}(\mathbf{x}_n, \mathbf{v}_i) = - \sum_{c=1}^{C} p(\hat{y} = c | \mathbf{X}_n, \mathbf{v}_i) \log p(\hat{y} = c | \mathbf{X}_n, \mathbf{v}_i), \qquad (2)$$

where $p(\hat{y} = c | \mathbf{X}_n, \mathbf{v}_i)$ denotes the averaged prediction across the selected augmentations similar to Eq. (1). In operation, we use the entropy of the initial prompt $\mathbf{v}_0$ as the threshold and select the online prompts with lower entropy, where the selected prompts are more confident on the test sample (Niu et al., 2022). Formally, we have the entropy-selected online prompts subset $\mathcal{E}_n$:

$$\mathcal{E}_n = \big\{ \mathbf{v}_i \in \mathcal{V}_n \mid \mathcal{D}_{ent}(\mathbf{x}_n, \mathbf{v}_i) \leq \mathcal{D}_{ent}(\mathbf{x}_n, \mathbf{v}_0) \big\}. \qquad (3)$$

Therefore, $\mathcal{E}_n$ contains the online prompts that produce more confident predictions than the initial prompt, indicating they incorporate more relevant information and are better suited for $\mathbf{x}_n$.

**Probability difference selection.** However, the entropy is not always reliable in test-time tuning, especially when encountering distribution shifts (Lee et al., 2024). When continually selected and optimized for low entropy, the online prompts can be tuned overconfidently and produce incorrect predictions with very low entropy, causing prompt collapse discussed in Section 3. To avoid selecting overconfident prompts, we further introduce a probability difference metric for dynamic prompt selection. The probability difference $\mathcal{D}_{pro}(\mathbf{x}_n, \mathbf{v}_i)$ quantifies the prediction probability differences between the original test sample $\mathbf{x}_n$ and its augmentations $\mathbf{X}_n$, assessing the sensitivity of the prompts to the changes in the structure information of the input sample.

Given a test sample $\mathbf{x}_n$ and online prompt $\mathbf{v}_i \in \mathcal{V}_n$, we calculate the probability difference as:

$$\mathcal{D}_{pro}(\mathbf{x}_n, \mathbf{v}_i) = p(\hat{y} = c^* | \mathbf{x}_n, \mathbf{v}_i) - p(\hat{y} = c^* | \mathbf{X}_n, \mathbf{v}_i), \qquad (4)$$

where $c^* = \arg\max_c p(\hat{y} = c | \mathbf{x}_n, \mathbf{v}_i)$ denotes the pseudo-label of the test sample $\mathbf{x}_n$ predicted by prompt $\mathbf{v}_i$. Prompts with higher $\mathcal{D}_{pro}$ are more sensitive to the changes of the sample $\mathbf{x}_n$, which are less likely to be overconfident. By contrast, lower $\mathcal{D}_{pro}$ indicates similar predictions regardless of input modifications, increasing the risk of overconfidence and prompt collapse, especially with low prediction entropy.

To circumvent overconfident prompts during dynamic selection, we propose to select the online prompts with higher $\mathcal{D}_{pro}$ values. Similar to the entropy metric, we use the probability difference of the initial prompt as the threshold for prompt selection. Formally, the subset of online prompts with high probability difference on the test sample $\mathbf{x}_n$ are formulated as:

$$\mathcal{R}_n = \big\{ \mathbf{v}_i \in \mathcal{V}_n \mid \mathcal{D}_{pro}(\mathbf{x}_n, \mathbf{v}_i) \geq \mathcal{D}_{pro}(\mathbf{x}_n, \mathbf{v}_0) \big\}. \qquad (5)$$

**Dynamic selected prompts.** Combining Eq. (3) and Eq. (5) together, we obtain the subset of selected prompt $\mathcal{S}_n$, where the selected prompt meets the requirements in both above selection processes:

$$\mathcal{S}_n = \mathcal{E}_n \cap \mathcal{R}_n. \qquad (6)$$

By taking the intersection of the two subsets in Eq. (6), the selected prompts simultaneously satisfy both lower entropy in Eq. (3) and larger probability differences in Eq. (5), which produce more confident predictions and are more sensitive to the changes of the test sample. Therefore, the selected prompts are more relevant to the test samples and have low risks of collapse. Moreover, since we utilize the entropy and probability differences of the initial prompt as thresholds, our method enables adaptively prompt selection for each test sample. By autonomously selecting relevant prompts for each test sample, the online prompts are enriched with more specific data distribution information, enhancing both the predictive performance of the current test sample and the optimization of the selected prompts. Since the irrelevant and collapse prompts are frozen, potential conflicting optimization directions for these prompts are avoided, thereby reducing error accumulation.

Note that the predictions and entropy in Eq. (2) and (4) are inherently calculated for test-time prompt tuning in Eq. (1). Thus, our method introduces very few extra operations for prompt selection.

**Optimization and prediction.** For each test sample $\mathbf{x}_n$, after dynamically selecting the online prompts $\mathcal{V}_n$, we first optimize the selected prompts by minimizing the entropy as:

$$\mathcal{L}_{ent}(\mathcal{S}_n; \mathbf{x}_n) = -\sum_{c=1}^{C} p(\hat{y} = c|\mathbf{X}_n, \mathcal{S}_n) \log p(\hat{y} = c|\mathbf{X}_n, \mathcal{S}_n), \ \ \widetilde{\mathcal{S}}_n \leftarrow \mathcal{S}_n - \alpha \nabla \mathcal{L}_{ent}(\mathbf{x}_n, \mathcal{S}_n), \ (7)$$

where $p(\hat{y} = c|\mathbf{X}_n, \mathcal{S}_n)$ denotes the average prediction probabilities across the selected prompts and sample augmentations. With the updated prompts $\widetilde{\mathcal{S}}_n$, we perform prediction for the test sample $\mathbf{x}_n$ as $\arg\max_c p(\hat{y} = c|\mathbf{x}_n, \widetilde{\mathcal{S}}_n)$.

## 4.2 DYNAMIC PROMPT APPENDING

During dynamic prompt selection, there can be no appropriate prompt in the prompt set $\mathcal{V}_n$ for specific test samples, leading to an empty $\mathcal{S}_n$. That means the online prompts in $\mathcal{V}_n$ are either irrelevant for the current sample or collapsed. In this case, utilizing existing online prompts for the sample can lead to conflict optimization or severe error accumulation. To fix this issue, we introduce dynamic prompt appending as a complementary strategy. Specifically, our method appends an initial prompt $\mathbf{v}_0$ into the prompt set $\mathcal{S}_n$ when it is empty. As a result, the selected prompt set $\mathcal{S}_n$ is reformulated as:

$$\mathcal{S}_n = \begin{cases} \{\mathbf{v}_0\}, & \text{if } \mathcal{E}_n \cap \mathcal{R}_n = \varnothing; \\ \mathcal{E}_n \cap \mathcal{R}_n, & \text{otherwise.} \end{cases} \quad (8)$$

The prompt in $\mathcal{S}_n$ is optimized the same as Eq. (7) and then utilized in the inference of the sample.

However, the size of the prompt buffer $\mathcal{V}_n$ cannot be infinite when appending new learnable prompts due to memory constraints and computational costs. To avoid infinitely increasing numbers of prompts in the prompt buffer, we set the maximum number $M$ of prompts in $\mathcal{V}_n$ as a hyperparameter, i.e., $M_n \leq M$ and introduce a prompt deletion mechanism: when a new prompt is appended into the prompt buffer and the current number of prompts is $M$, the method will remove the most inactive prompt $\mathbf{v}_{\text{inactive}}$ from the buffer. By optimizing $\mathcal{S}_n$ in Eq. (8) to $\widetilde{\mathcal{S}}_n$ through entropy minimization in Eq. (7), we formulate the update of the prompt buffer with dynamic prompt appending as:

$$\mathcal{V}_{n+1} = \begin{cases} \mathcal{V}_n + \widetilde{\mathcal{S}}_n - \{\mathbf{v}_{\text{inactive}}\}, & \text{if } \mathcal{E}_n \cap \mathcal{R}_n = \varnothing \ \text{ and } \ M_n = M; \\ \mathcal{V}_n + \widetilde{\mathcal{S}}_n - \mathcal{S}_n, & \text{otherwise.} \end{cases} \quad (9)$$

It is worth noting that the "+" and "−" operations are the append and delete operations for the prompt buffer. As shown in Figure 3, we always put the optimized prompts in $\widetilde{\mathcal{S}}_n$ at the start of the prompt buffer. Therefore, we achieve the deletion mechanism by entirely removing the online prompt at the end of the buffer, which has not been activated for the maximum allowed time. By appending new online prompts and ejecting the inactive ones, our dynamic prompt tuning effectively incorporates information from new data distributions and reduces error accumulation. We provide an algorithm of our method in Appendix A.

## 5 RELATED WORK

**Prompt learning.** To adapt vision-language models such as CLIP (Radford et al., 2021) and ALIGN (Jia et al., 2021) to downstream tasks, prompt learning methods are introduced (Lester et al., 2021; Li & Liang, 2021; Zhou et al., 2022b). Zhou et al. (2022b) propose learnable prompts in the input embedding space of the language model in CLIP. ProGrad (Zhu et al., 2023) aligns the gradients of the learnable prompts with the original prompt. In addition to the language input space, Bahng et al. (2022) introduces prompt learning into the vision branch of the CLIP model. Khattak et al. (2023) further proposes joint prompts for both vision and language encoders. To improve the generalization ability of the learned prompts, Zhou et al. (2022a) introduce imaging conditions into the language prompts. KgCoOp (Yao et al., 2023) reduces the forgetting of the general knowledge in the CLIP model by reducing the discrepancy between the learnable and handcrafted prompts. Derakhshani et al. (2023) propose Bayesian prompt learning to incorporate uncertainty in the learnable prompts. CoPrompt (Roy & Etemad, 2024) enforces the prediction consistency of the trainable and pre-trained models to prevent overfitting on the downstream task. Any-shift prompting (Xiao et al., 2024)

generates the test-specific prompt for visual and text encoders in a single feedforward pass, without fine-tuning at test time. Our method also aims to address distribution shifts in downstream tasks adaptation of the CLIP model. Differently, we tune the prompt at test time to adapt the prompts to unseen distributions.

**Test-time adaptation.** Test-time adaptation (Liang et al., 2023; Xiao & Snoek, 2024) aims to adapt pretrained models to test data distributions at test time by unsupervised optimization objectives. The idea of optimizing models at test time was first proposed by Sun et al. (2020), who introduced auxiliary self-supervised objective functions for test time training, followed by several recent works (Liu et al., 2021; Hardt & Sun, 2024). Alternatively, Wang et al. (2021) proposed model adaptation by entropy minimization at test time, which is widely used and investigated in recent test-time adaptation algorithms (Zhang et al., 2022; Goyal et al., 2022; Niu et al., 2022; 2023). There are two main settings for test-time adaptation methods, online adaptation (Wang et al., 2021; Iwasawa et al., 2021; Lee et al., 2023; Zhang et al., 2023; 2024b) and batch-wise adaptation (Schneider et al., 2020; Gao et al., 2022; Lim et al., 2023). Online adaptation incrementally improves the adaptation by gradually incorporating test information from online samples, while taking the risk of error accumulation (Wang et al., 2021; Niu et al., 2022; Lee et al., 2023). Batch-wise adaptation adapts the pretrained model to each test batch individually, avoiding error accumulation of online learning while ignoring the context information (Xiao et al., 2022; Gao et al., 2022). Our method introduces online optimization into test-time prompt tuning while introducing a dynamic tuning method to reduce error accumulation.

**Test-time prompt tuning.** To address distribution shifts during the downstream task adaptation of the CLIP model, recent works propose test-time prompt tuning. TPT (Shu et al., 2022) tunes the prompts in the text embedding space of the CLIP model at test-time, using an entropy minimization objective on randomly augmented samples. Following TPT, DiffTPT (Feng et al., 2023) generates more data for test-time tuning by pretrained diffusion models. Samadh et al. (2023) test-time tunes the prompts pretrained by MaPLe (Khattak et al., 2023) by aligning the test sample statistics to the offline source statistics. C-TPT (Yoon et al., 2024) considers the calibration error for tuning the prompts. RLCF (Zhao et al., 2024) adopts an extra CLIP model as the reward model to provide feedback during test-time prompt tuning. Following TPT, most previous methods independently tune the prompt for each test sample. AdaPrompt (Zhang et al., 2024a) performs prompt tuning on each batch of 64 test samples with a buffer to store confident, class-balanced samples for improved tuning and prediction. Our method aims to utilize the beneficial information from online samples to enhance the test-time prompt tuning for each individual sample while reducing error accumulation.

## 6 EXPERIMENTS

**Fifteen datasets.** Following previous methods (Shu et al., 2022; Samadh et al., 2023), we conduct experiments across two settings that suffer from distribution shifts to demonstrate the effectiveness of our method: domain generalization and cross-dataset shifts. For the domain generalization setting, we evaluate the method on ImageNet (Deng et al., 2009) and its four variant datasets: ImageNet-V2 (Recht et al., 2019), ImageNet-(S)ketch (Wang et al., 2019), ImageNet-A (Hendrycks et al., 2021b), and ImageNet-R (Hendrycks et al., 2021a). For the cross-dataset setting, we evaluate our method on 10 image classification datasets covering various tasks: Caltech101 (Fei-Fei et al., 2004), OxfordPets (Parkhi et al., 2012), StanfordCars (Krause et al., 2013), Flowers102 (Nilsback & Zisserman, 2008), Food101 (Bossard et al., 2014), FGVCAircraft (Maji et al., 2013), SUN397 (Xiao et al., 2010), DTD (Cimpoi et al., 2014), EuroSAT (Helber et al., 2019), and UCF101 (Soomro et al., 2012).

**Implementation details.** Based on the CLIP model with ViT-Base-16 (Dosovitskiy et al., 2020), we initialize our dynamic prompts with the manually crafted "a photo of a" and optimize the prompts online in the text input embedding space. The prompt set optimized by one test sample is utilized for the next sample. Following TPT (Shu et al., 2022), we generate 63 augmentations by random resize crops for each individual test image to construct a batch of 64 images including the original image. During the dynamic tuning, we calculate the entropy and augmentation probability differences over these 63 augmented images as the dynamic prompt selection metrics. The thresholds are obtained in the same way based on the initial prompt. We set the maximum number of the prompt set $M$ as 10. We append new prompts to the dynamic prompt set when no appropriate prompt is selected for the test sample. Once the number of prompts in the prompt set $\mathcal{V}$ exceeds $M$, we remove the prompt that has been inactive for the longest time. For optimization, we select the top 10% confident samples

Table 1: **Comparisons on the domain generalization setting** with both prompt learning and test-time prompt tuning methods. The prompt learning methods train their prompts on ImageNet. The proposed method outperforms both kinds of methods in terms of accuracy. Combining our method with pretrained prompts further improves the performance. Reproduced results are indicated in *italics*.

| Method | ImageNet | ImageNet-V2 | ImageNet-S | ImageNet-A | ImageNet-R | *OoD Mean* |
|---|---|---|---|---|---|---|
| CLIP (Radford et al., 2021) | 66.73 | 60.86 | 46.09 | 47.87 | 73.98 | 57.20 |
| *Prompt learning methods without test-time tuning* | | | | | | |
| CoOp (Zhou et al., 2022b) | 71.51 | 64.20 | 47.99 | 49.71 | 75.21 | 59.28 |
| CoCoOp (Zhou et al., 2022a) | 71.02 | 64.07 | 48.75 | 50.63 | 76.18 | 59.90 |
| KgCoOp (Yao et al., 2023) | 71.20 | 64.10 | 48.97 | 50.69 | 76.70 | 60.11 |
| MaPLe (Khattak et al., 2023) | 70.72 | 64.07 | 49.15 | 50.90 | 76.98 | 60.28 |
| CoPrompt (Roy & Etemad, 2024) | 70.80 | 64.25 | 49.43 | 50.50 | 77.51 | 60.42 |
| Any-shift Prompt (Xiao et al., 2024) | - | 64.53 | 49.80 | 51.52 | 77.56 | 60.85 |
| *Test-time prompt tuning methods* | | | | | | |
| TPT (Shu et al., 2022) | 68.98 | 63.45 | 47.94 | 54.77 | 77.06 | 60.81 |
| AdaPrompt (Zhang et al., 2024a) | - | *59.32* | *47.72* | *47.71* | 73.98 | 57.18 |
| DiffTPT (Feng et al., 2023) | 70.30 | 65.10 | 46.80 | 55.68 | 75.00 | 60.65 |
| C-TPT (Yoon et al., 2024) | 69.30 | 63.40 | 48.50 | 52.90 | 78.00 | 60.70 |
| ***This paper*** | 69.61 | 64.67 | 48.22 | 56.17 | 78.17 | 61.81 |
| CoOp + TPT (Shu et al., 2022) | 73.61 | 66.83 | 49.29 | 57.95 | 77.27 | 62.84 |
| CoOp + ***This paper*** | **74.08** | **67.25** | **50.28** | 60.55 | 79.15 | **64.31** |
| MaPLe + TPT (Shu et al., 2022) | 71.87 | 64.87 | 48.16 | 58.08 | 78.12 | 62.31 |
| MaPLe + PromptAlign (Samadh et al., 2023) | - | 65.29 | 50.23 | 59.37 | 79.33 | 63.56 |
| MaPLe + ***This paper*** | 72.71 | 66.34 | 50.25 | **60.72** | **79.57** | 64.22 |

among the batch and calculate the entropy of the averaged logits of the selected predictions following Shu et al. (2022). We utilize a learning rate of $0.005$ for domain generalization and $0.003$ for the cross-dataset settings with the AdamW optimizer.

When combing our method with other prompt tuning methods (Zhou et al., 2022b; Khattak et al., 2023), we initialize the prompts as the pretrained prompts of these methods trained on ImageNet. When integrated with MaPLe (Khattak et al., 2023), our dynamic test-time prompt tuning is applied on *both* the textual and visual branches, therefore evolved into a multimodal setting. Our method runs on an NVIDIA A100 GPU.

## 6.1 COMPARISIONS

**Comparisons on domain generalization setting.** We compare our method on the domain generalization setting with both prompt learning (Zhou et al., 2022a;b; Khattak et al., 2023; Roy & Etemad, 2024) and test-time prompt tuning methods (Shu et al., 2022; Samadh et al., 2023). The prompt learning methods train their prompts by supervised cross-entropy loss on ImageNet. As shown in Table 1, our method achieves better overall performance compared with the prompt learning methods. Moreover, since our DynaPrompt is orthogonal to most of these prompt learning methods, applying our method together with prompt learning methods like CoOp (Zhou et al., 2022b) and MaPLe (Khattak et al., 2023)) further improves the performance.

DynaPrompt also surpasses recent test-time prompt tuning methods. Our method outperforms TPT (Shu et al., 2022) on all datasets with both hand-crafted and learned prompts (Zhou et al., 2022b; Khattak et al., 2023), achieving at least $1\%$ improvements. The method also achieves better overall performance compared with other recent methods DiffTPT (Feng et al., 2023), AdaPrompt (Zhang et al., 2024a), and C-TPT (Yoon et al., 2024). Compared with PromptAlign (Samadh et al., 2023), which is constructed on MaPLe (Khattak et al., 2023) and utilizes extra source data during test-time tuning, our method is also superior.

**Comparisons on cross-dataset setting.** On the cross-dataset setting, we also compare our method with both prompt learning and test-time prompt tuning methods. As shown in Table 2, our method outperforms the common prompt learning methods for 8 of the 10 datasets and achieves the best overall performance. Compared with the test-time prompt tuning methods, the proposed method again outperforms TPT with both hand-crafted and learned prompts (Khattak et al., 2023). Our method also achieves higher accuracy compared with the recent test-time prompt tuning methods Samadh et al. (2023). We observe that the improvements on these datasets for our method, and even most test-time prompt tuning methods, are not as obvious as in the domain generalization setting. The

Table 2: **Comparisons on the cross-dataset setting** with both prompt learning and test-time prompt tuning methods. The prompt learning methods train their prompts on ImageNet. The proposed method again outperforms both kinds of methods on 6 datasets and achieves the best overall performance. Reproduced results are indicated in *italics*.

| Method | Caltech | Pets | Cars | Flowers | Food101 | Aircraft | SUN397 | DTD | EuroSAT | UCF101 | Mean |
|---|---|---|---|---|---|---|---|---|---|---|---|
| CLIP | 93.35 | 88.25 | 65.48 | 67.44 | 83.65 | 23.67 | 62.59 | 44.27 | 42.01 | 65.13 | 63.58 |
| *Prompt learning methods without test-time tuning* | | | | | | | | | | | |
| CoOp | 93.70 | 89.14 | 64.51 | 68.71 | 85.30 | 18.47 | 64.15 | 41.92 | 46.39 | 66.55 | 63.88 |
| CoCoOp | 94.43 | 90.14 | 65.32 | 71.88 | 86.06 | 22.94 | 67.36 | 45.73 | 45.37 | 68.21 | 65.74 |
| MaPLe | 93.53 | 90.49 | 65.57 | 72.23 | 86.20 | 24.74 | 67.01 | 46.49 | 48.06 | 68.69 | 66.30 |
| CoPrompt | 94.50 | 90.73 | 65.67 | 72.30 | 86.43 | 24.00 | 67.57 | 47.07 | **51.90** | 69.73 | 67.00 |
| *Test-time prompt tuning methods* | | | | | | | | | | | |
| TPT | 94.16 | 87.79 | 66.87 | 68.98 | 84.67 | 24.78 | 65.50 | 47.75 | 42.44 | 68.04 | 65.10 |
| DiffTPT | 92.49 | 88.22 | 67.01 | 70.10 | **87.23** | **25.60** | 65.74 | 47.00 | 41.04 | 68.22 | 65.47 |
| C-TPT | 93.60 | 88.20 | 65.80 | 69.80 | 83.70 | 24.00 | 64.80 | 46.00 | 43.20 | 65.70 | 64.80 |
| AdaPrompt | 94.07 | 89.64 | *63.29* | 72.97 | 84.72 | *21.21* | *65.37* | 44.75 | 47.20 | 67.22 | 65.04 |
| ***This paper*** | 94.32 | 88.28 | 67.65 | 69.95 | 85.42 | 24.33 | 66.32 | 47.96 | 42.28 | 68.72 | 65.52 |
| CoOp + TPT | 93.15 | 89.48 | 66.77 | 68.48 | 86.48 | 20.51 | 66.06 | 43.32 | 37.73 | 68.91 | 64.09 |
| CoOp + ***This paper*** | 94.40 | 90.04 | 67.35 | 69.38 | 86.45 | 21.35 | 66.17 | 46.98 | 38.55 | 69.54 | 65.02 |
| MaPLe + TPT | 93.59 | 90.72 | 66.50 | 72.37 | 86.64 | 24.70 | 67.54 | 45.87 | 47.80 | 69.19 | 66.49 |
| MaPLe + PromptAlign | 94.01 | 90.76 | **68.50** | 72.39 | 86.65 | 24.80 | 67.54 | 47.24 | 47.86 | 69.47 | 66.92 |
| MaPLe + ***This paper*** | **95.17** | **90.95** | 68.26 | **73.28** | 86.60 | 24.36 | **68.18** | **48.75** | 47.53 | **69.85** | **67.29** |

Table 3: **Ablations on our dynamic prompt selection strategy.** Either without the entropy-based selection in Eq. (3) or the probability difference selection in Eq. (5) leads to obvious performance degradation.

| Method | ImageNet-V2 | ImageNet-S | ImageNet-A | ImageNet-R | *Mean* |
|---|---|---|---|---|---|
| w/o entropy | 61.45 | 46.86 | 50.25 | 76.19 | 58.69 |
| w/o probability difference | 61.93 | 46.95 | 50.95 | 77.37 | 59.23 |
| With both selection metrics | 64.67 | 48.22 | 56.17 | 78.17 | 61.81 |

Table 4: **Ablations on dynamic prompt appending strategy.** Without the appending strategy, the performance drops considerably due to error accumulation and prompt collapse.

| Method | ImageNet-V2 | ImageNet-S | ImageNet-A | ImageNet-R | *Mean* |
|---|---|---|---|---|---|
| w/o dynamic appending | 35.66 | 20.38 | 30.74 | 43.72 | 32.63 |
| w/ dynamic appending | 64.67 | 48.22 | 56.17 | 78.17 | 61.81 |

main reason can be that the fine-grained tasks (e.g., Aircraft and Food101) and specific tasks (e.g., EuroSAT with satellite images) are more detailed and challenging for prompt tuning at test-time.

## 6.2 ABLATION STUDIES

**Effectiveness of dynamic prompt selection and appending.** To investigate the roles of our dynamic selection and appending strategies of dynamic prompt learning, we conduct experiments on domain generalization datasets. As shown in Table 3, removing either the entropy or the probability difference metric during dynamic prompt selection results in obvious performance degradation on all datasets. Without the entropy, the selected prompts can be either confident or uncertain to the test sample, which is not suitable for the sample, leading to performance degradation. Without the probability difference, the proposed method may select collapsed prompts during test-time tuning, resulting in the wrong direction of the prediction and optimization.

As shown in Table 4, the performance degrades considerably when we remove the dynamic prompt appending strategy from our method. Without the appending and removing strategy, the method can only select the online prompts in the buffer, even if there is no appropriate one. In this case, the risks of optimization in conflict directions are highly amplified. The error during the optimization is then accumulated in the sequential online samples, leading to prompt collapse.

**Analyses on error accumulation.** Here we provide more analysis on our DynaPrompt. As shown in Table 5, TPT achieves good overall performance as it avoids error accumulation by independent prompt tuning. Online TPT performs competitively at the start but declines rapidly as shown in Figure 2. While Oracle also has a performance degradation initially, the performance

Table 5: **Analyses on error accumulation.** DynaPrompt reduces error accumulation and benefits from selected online prompts, outperforming TPT and OnlineTPT.

| Method | TPT | Online TPT | Oracle | *This paper* |
|--------|-------|------------|--------|--------------|
| Accuracy | 54.77 | 6.96 | 59.38 | 56.17 |

stabilizes since it avoids error accumulation by tuning prompts only with correct predictions. By incorporating the beneficial information from online samples, Oracle surpasses TPT and Online TPT. Our DynaPrompt reduces error accumulation and achieves stable performance by dynamically selecting, appending, and deleting the online prompts. Therefore, the method achieves good overall performance and beats TPT by incorporating relevant information in online data. Moreover, our method performs better during online learning, proving its capability to capture beneficial information.

**Influence of the prompt buffer size.** We also ablte the influence of the prompt buffer size $M$ on our method in Figure 4a. The experiments are conducted on ImageNet-A. As the prompt buffer size increases, the proposed method shows an upward slope. The improvement is faster when the buffer size is smaller than 10. Figure 4b shows the time costs of the proposed method, which continually increases with larger buffer sizes. Compared with TPT, our method requires more time costs, which can be a limitation of the approach. Note that most of the additional

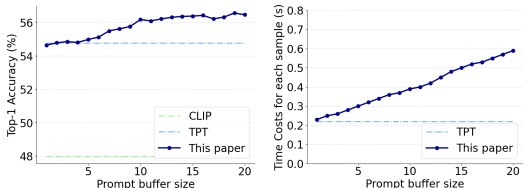

(a) Top-1 Accuracy of different prompt buffer sizes. (b) Time costs with different prompt buffer sizes.

Figure 4: **Influence of the prompt buffer size** on performance (a) and time costs (b). Larger buffer sizes lead to better performance with higher costs.

time cost stems from optimizing multiple prompts rather than the selection and appending strategy. For instance, with the buffer size 10, the total processing time per test sample is approximately 0.39 seconds, of which the selection and appending steps account for only 0.004 seconds. For a good trade-off between performance and time costs, we set the maximum size of the prompt buffer to 10.

**Sensitivity to test time sample order.** As our DynaPrompt optimizes the prompt online, the performance of the method can be influenced by the order of the test samples. To investigate this influence, we conduct experiments on ImageNet-A and ImageNet-R for six rounds, which have different sample orders. We shuffle the sample order with different random seeds at test time, which leads to different sample orders. The results are provided in Figure 5a and 5b, respectively. Order 0 denotes the default order the same as the experiments of TPT (Shu et al.,

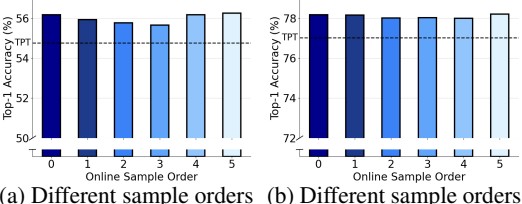

(a) Different sample orders for ImageNet-A. (b) Different sample orders for ImageNet-R.

Figure 5: **Sensitivity to test-time sample order** on ImageNet-A (a) and ImageNet-R (b). More test samples lead to more stable performance.

2022). We observe that there are fluctuations in the performance of both datasets. The performance on ImageNet-R is more stable with a larger number of test samples (30,000) than ImageNet-A (7,500). Nevertheless, independent of the test order, the proposed method surpasses TPT consistently.

## 7 CONCLUSION

In this paper, we propose DynaPrompt, a new test-time prompt tuning approach, which exploits beneficial information from the online test samples while alleviating error accumulation. Our method introduces a dynamic prompt buffer, which adaptively selects and optimizes prompts for each test sample. The selected prompts incorporate relevant information from previous test samples, thereby benefiting the prediction for the current sample. By optimizing the selected prompts while freezing the rest, the method further enhances the learned prompts to incorporate relevant information from test data. DynaPrompt also enables the buffer to autonomously append new learnable prompts and delete the inactive ones, improving adaptability to new test data and reducing the risk of error accumulation. Experiments on fourteen benchmarks validate the effectiveness of our proposal.

## REPRODUCIBILITY

We include all necessary details to facilitate the reproducibility of our work. The experimental setup, including benchmarks, model configurations, hyperparameters, and evaluation protocols, is thoroughly explained in the experiments section. We also give an algorithm in the Appendix to provide the detailed process of our method. We will make our code publicly available.

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

# A    ALGORITHM

---

**Algorithm 1** Dynamic Test-Time Prompt Tuning (DynaPrompt)

---

1: **Input:** Test samples $\{\mathbf{x}_n\}_{n=0}^{N}$; a prompt buffer $\mathcal{V}_n$ with length $M_n$, initialized as $\mathcal{V}_0 = \varnothing$; maximum buffer size $M$; hand-crafted or pretrained initial prompt $\mathbf{v}_0$
2: **for** $n = 0 : N$ **do**
3:     Randomly augment $\mathbf{x}_n$ to $\mathbf{X}_n$.
4:     Obtain predictions $\{p(y|\mathbf{X}_n, \mathbf{v}_i)\}_{i=1}^{M_n}$ and $p(y|\mathbf{X}_n, \mathbf{v}_0)$.

      `// Dynamic prompt selection.`
5:     Calculate $\mathcal{D}_{ent}(\mathbf{x}_n, \mathbf{v}_i)$ and $\mathcal{D}_{ent}(\mathbf{x}_n, \mathbf{v}_0)$ by Eq. (2), then select prompts subset $\mathcal{E}_n$ by Eq. (3).
6:     Calculate $\mathcal{D}_{pro}(\mathbf{x}_n, \mathbf{v}_i)$ and $\mathcal{D}_{pro}(\mathbf{x}_n, \mathbf{v}_0)$ by Eq. (4), then select prompts subset $\mathcal{R}_n$ by Eq. (5).
7:     Select relevant prompts $\mathcal{S}_n$ by $\mathcal{E}_n \cap \mathcal{R}_n$.

      `// Dynamic prompt appending.`
8:     **if** $\mathcal{S}_n = \varnothing$ **then**
9:         $\mathcal{S}_n = \{\mathbf{v}_0\}$.
10:    **end if**

      `// Optimizing selected prompts.`
11:    Tune the prompts in $\mathcal{S}_n$ by entropy minimization in Eq. (7): $\widetilde{\mathcal{S}}_n \leftarrow \mathcal{S}_n - \alpha \nabla \mathcal{L}_{ent}(\mathbf{x}_n, \mathcal{S}_n)$.

      `// Update the prompt buffer` $\mathcal{V}_n$ `to` $\mathcal{V}_{n+1}$`.`
12:    **if** $M_n = M$ and $\mathcal{E}_n \cap \mathcal{R}_n = \varnothing$ **then**
13:        Append the updated prompt $\widetilde{\mathcal{S}}_n$ to the top of $\mathcal{V}_n$.
14:        Remove the prompt $\mathbf{v}_{inactive}$ at the bottom of $\mathcal{V}_n$.
15:    **else**
16:        Append the optimized prompts in $\widetilde{\mathcal{S}}_n$ to the top of $\mathcal{V}_n$.
17:        Remove the selected prompts in $\mathcal{S}_n$ from $\mathcal{V}_n$.
18:    **end if**
19: **end for**

---

# B    DETAILED IMPLEMENTATIONS

**Details of data augmentation.** To generate the augmented data $\mathbf{X}_n$ for each sample, we follow the same data augmentation strategy used in TPT (Shu et al., 2022). That is, we use AugMix (Hendrycks et al., 2020) to augment the original test image into 63 different augmentation samples, leading to 64 samples in total for each test image. Each test image is first augmented by *resize* and random *crop*, then fed into the AugMix strategy with several augmentation methods including *auto contrast, equalization, posterization, rotation, solarization, shearing, and translating*.

# C    EXTRA EXPERIMENTS

**Effect of initial prompts.** The initial prompts can affect the final performance in prompt learning (Zhou et al., 2022a). To investigate the effect of the initial prompts of the proposed method, we conducted experiments on ImageNet-A using various initial text prompts. As shown in Table 6, the initial prompts affect the performance of CLIP (Radford et al., 2021), TPT (Shu et al., 2022), as well as our method. The reason can be related to the initial predictions of the original CLIP model. Nonetheless, our method consistently outperforms TPT, showing robustness despite variations in initialization.

**Ablations on prompt length for online test-time prompt tuning.** To investigate the prompt length effect, we experiment with longer prompts for both TPT and Online TPT. We set the prompt length to 40, which is 10 times longer than the 4-item original "a photo of a". We consider two types of long prompts: (A) 10 times copy of *"a photo of a"*, (B) *"Let us solve an image classification task: a photo of a distinct object, animal, plant, or scene, captured in diverse environments and representing meaningful categories. Carefully analyze its features; the exact category of the photo is a"*, generated by GPT-4o.

Table 6: **Effect of different initial prompts.** The initial prompts affect the performance of CLIP (Radford et al., 2021), TPT (Shu et al., 2022), as well as our method. Our method consistently outperforms TPT, showing robustness despite variations in initialization.

| Initial prompt | CLIP | TPT | *This paper* |
|---|---|---|---|
| "*a photo of a*" | 47.87 | 54.77 | **56.17** |
| "*an image of a*" | 48.31 | 54.84 | **56.19** |
| "*high-quality of a*" | 44.48 | 51.47 | **52.57** |
| "*Identify feature of*" | 46.08 | 50.18 | **51.84** |
| "*Visual features of*" the | 46.21 | 51.48 | **53.13** |
| "*natural photo of a*" | 44.82 | 52.56 | **54.08** |
| "*classify the photo of*" | 46.60 | 52.51 | **53.27** |
| *Mean* | 46.35 | 52.54 | **53.89** |

Table 7: **Ablations on prompt length for online test-time prompt tuning.** Well-designed long prompt improves the performance of the original CLIP, but may lead to more difficult optimization for test-time prompt tuning, resulting in worse TPT performance. The long prompts may even lead to worse prompt collapse due to the difficult optimization and error accumulation. By contrast, our method achieves better performance while reducing error accumulation by our dynamic prompt selection and appending strategies.

| Method | Initial prompt | Trainable prompt length | Accuracy |
|---|---|---|---|
| CLIP | "*a photo of a*" | 4 | 47.87 |
| | long prompt (A) | 40 | 46.99 |
| | long prompt (B) | 40 | 48.21 |
| TPT | "*a photo of a*" | 4 | 54.77 |
| | long prompt (A) | 40 | 52.97 |
| | long prompt (B) | 40 | 52.23 |
| Online TPT | "*a photo of a*" | 4 | 6.96 |
| | long prompt (A) | 40 | 2.06 |
| | long prompt (B) | 40 | 4.24 |
| *This paper* | "*a photo of a*" | 4 * 10 | **56.17** |

As shown in Table 7, longer trainable prompts do not solve the problem of prompt collapse for online learning, even worsening the problem. Since the online testing fails due to error accumulation and prompt collapse, simply improving the length of the prompts does not help. Specifically designed long prompts (B) perform better on the optimization-free CLIP model. However, it may lead to more difficult optimization for test-time tuning, resulting in worse TPT performance. By contrast, based on the prompt selection and appending strategy, our method achieves better performance while reducing error accumulation and prompt collapse.

**Experiments on different backbones.** To evaluate the proposed method on different backbones, we conduct experiments for ImageNet-based datasets on ResNet-50 and ViT-B/32. The experiments are provided in Table 8. Our method again outperforms CLIP and TPT.

**Online prompts tuning with identical initialization.** To provide more insights into the prompt collapse of online prompt tuning, we conduct experiments on ImageNet-A with 10 initialized online prompts. The prompts are initialized by the embedding of "a photo of a" with random noise to be identical. We use our selection strategy to select and optimize the prompts online. When no prompt is selected, we use the initial prompt "a photo of a" for prediction.

As shown in Table 9, the variant method outperforms online TPT and CLIP, which demonstrates that it reduces collapse since it provides diverse optimization directions for different test samples through dynamic selection. However, the method underperforms TPT and our DynaPrompt. However, it underperforms TPT and our method. The reason can be that the negative influence of previous test

Table 8: **Experiments on different backbones.** Our method again outperforms CLIP (Radford et al., 2021) and TPT (Shu et al., 2022), despite the backbone of the CLIP model.

| Method | ImageNet | ImageNet-V2 | ImageNet-S | ImageNet-A | ImageNet-R | mean | OoD mean |
|---|---|---|---|---|---|---|---|
| ResNet-50 | | | | | | | |
| CLIP | 58.16 | 51.41 | 33.37 | 21.83 | 56.15 | 44.18 | 40.69 |
| TPT | 60.74 | 54.7 | 35.09 | 26.67 | 59.11 | 47.26 | 43.89 |
| *This paper* | **61.56** | **55.12** | **35.64** | **27.84** | **60.63** | **48.16** | **44.81** |
| ViT-B/32 | | | | | | | |
| CLIP | 62.05 | 54.79 | 40.82 | 29.57 | 65.99 | 50.64 | 47.79 |
| TPT | 63.64 | 57.22 | 41.66 | 34.63 | 69.42 | 53.31 | 50.73 |
| *This paper* | **64.72** | **58.10** | **42.04** | **36.05** | **70.46** | **54.27** | **51.66** |

Table 9: **Identical initialization online prompts with dynamic selection.** We introduce 10 online prompts with identical initialization and dynamically select and optimize subsets of the prompts at test time. The method outperforms CLIP and OnlineTPT, demonstrating the variant method reduces prompt collapse during online tuning. However, the performance is still worse than TPT and Our DynaPrompt, indicating error accumulation still exists.

| Method | ImageNet-A |
|---|---|
| CLIP | 47.87 |
| Online TPT | 6.96 |
| Identical initialized online prompts with dynamic selection | 48.14 |
| TPT | 54.77 |
| *This paper* | **56.17** |

samples during online updating is still not entirely solved by the limited number of predefined online prompts, which leads to error accumulation and suboptimal predictions.

