# OpenReview forum: "DynaPrompt: Dynamic Test-Time Prompt Tuning"
_ICLR.cc/2025/Conference — ICLR 2025 Poster_

### Official Review · Reviewer_kMEy · 2024-11-01

**Soundness:** 3
**Presentation:** 3
**Contribution:** 3
**Rating:** 8
**Confidence:** 5

**Summary:**

This paper introduces DynaPrompt, a test-time prompt-tuning (TPT) approach that exploit information from previous test samples while avoiding error accumulation. While naively adapting TPT to the online setting lead to collapse, DynaPrompt leverages a dynamic buffer of prompts and optimize only the most relevant prompts for each test sample. Furthermore, DynaPrompt introduces only one additional hyper-parameter, the buffer size, thanks to an adaptative thresholding strategy.  DynaPrompt demonstrate consistent improvement over TPT and its variants (CoOp+TPT, MaPLe+TPT) making it a simple and effective alternative.

**Strengths:**

- **Motivation**: The authors made a good job motivating the paper, illustrating the potential benefit of moving to the on-line scenario while showing the non-triviality of extending TPT to this setup.
- **Contribution**: The technical contribution of the paper is simple and effectively solve a clearly identified issue.
- **Clarity**: The paper is easy to follow and arguments are clearly articulated.
- **Experiments**: The experiments are convincing and show consistent improvements over the baselines.

**Weaknesses:**

- **Missing experiments** : Some experiments are missing like evaluating on different backbones and more importantly evaluation on the Imagenet dataset. I will consider raising my score if results on the Imagenet dataset are added.

**Questions:**

- Unless I missed it, it seems that there is no results on the Imagenet dataset. Could you provide results on the Imagenet dataset ?
- Could you provide evaluation results on other CLIP vision backbones such as ViT-B/32 or Resnet-50 ?
- Why did you not included results for CoOp + DynaPrompt in Table 2 ?
- If I understand your method correctly, the buffer initially contains only one prompt initialized as ‘a photo of a’. For a given test sample, if your selection criteria (entropy and probability difference) are not met, a new prompt is initialized with ‘a photo of a’; otherwise, selected past prompts are used for optimization. I’m curious whether you considered initializing a set of prompts from the beginning and using your selection criteria to optimize subsets of this buffer. Initially, all prompts would be identical, but a small degree of randomness at the start could help address this issue. Do you think collapse would occur in that scenario? An experimental result would be helpful, but your thoughts or intuition on this would be sufficient.

---

> ### Author Response · Authors · 2024-11-20
> **Response to Reviewer kMEy**
>
> We thank Reviewer kMEy for the constructive feedback and insightful comments. We hope to address the concerns of the reviewer with the responses below.
>
>
> **Questions**
>
> **Experiments on ImageNet**
>
> We provide ImageNet results of our method, based on the ViT-B/16 backbone, in the following table, Our method achieves better performance than CLIP and TPT through dynamic prompt tuning.
> Combined with the prompt learning methods CoOp and MaPLe, the performance of our method is further improved. We added the ImageNet results in Table 1.
>
> | Method  | ImageNet |
> |--------------------|----------|
> | CLIP               | 66.73    |
> | CoOp               | 71.51    |
> | CoCoOp             | 71.02    |
> | MaPLe              | 70.72    |
> | TPT                | 68.98    |
> | ***This paper***         | 69.61    |
> | CoOp + TPT         | 73.61    |
> | CoOp + ***This paper***  | **74.08**    |
> | MaPLe + TPT        | 71.87    |
> | MaPLe + ***This paper*** | 72.71    |
>
>
>
>
> **Experiments with different backbones**
>
> To evaluate the proposed method on different backbones, we conduct experiments for ImageNet-based datasets with the ResNet-50 and ViT-B/32 backbones. The experiments are provided in the following table. No matter the backbone, our method outperforms CLIP and TPT. We added these results to Appendix C.
>
> | RN 50      | ImageNet | ImageNet-V2 | ImageNet-S | ImageNet-A | ImageNet-R | mean  | OoD mean |
> |------------|----------|-------------|------------|------------|------------|-------|----------|
> | CLIP       | 58.16    | 51.41       | 33.37      | 21.83      | 56.15      | 44.18 | 40.69    |
> | TPT        | 60.74    | 54.7        | 35.09      | 26.67      | 59.11      | 47.26 | 43.89    |
> | ***This paper*** | **61.56**    | **55.12**    | **35.64**   | **27.84**   | **60.63**   | **48.16** | **44.81**    |
>
>
> | ViT-B/32   | ImageNet | ImageNet-V2 | ImageNet-S | ImageNet-A | ImageNet-R | mean   | OoD mean |
> |------------|----------|-------------|------------|------------|------------|--------|----------|
> | CLIP     | 62.05    | 54.79       | 40.82      | 29.57      | 65.99      | 50.64 | 47.79  |
> | TPT      | 63.64    | 57.22       | 41.66      | 34.63      | 69.42      | 53.31 | 50.73  |
> | ***This paper*** |     **64.72**    | **58.10**       | **42.04**      | **36.05**      | **70.46**      |    **54.27**    | **51.66**    |
>
>
>
>
> **Results for  CoOp + DynaPrompt in the cross-dataset setting**
>
> We provide the requested results on the cross-dataset setting. Our method outperforms TPT based on the CoOp pretrained prompt for 9 out of 10 datasets. We included these results in Table 2.
>
> |  Method    | Caltech  | Pets  | Cars  | Flowers  | Food101  | Aircraft  | SUN397  | DTD   | EuroSAT | UCF101  | Average |
> |-------------------|----------|-------|-------|----------|----------|-----------|---------|-------|---------|---------|---------|
> | CoOp + TPT           | 93.15    | 89.48 | 66.77 | 68.48    | 86.48    | 20.51     | 66.06   | 43.32 | 37.73   | 68.91   | 64.09  |
> | CoOp + ***This paper*** | 94.40    | 90.04 | 67.35 | 69.38    | 86.45    | 21.35     | 66.17   | 46.98 | 38.55   | 69.54   | 65.02  |
>
>
> **Online prompts tuning with identical initialization**
>
> The reviewer is correct that our buffer is initialized with only one prompt, and we append a new prompt initialized with “a photo of a” when no previous prompt is selected during online tuning. To demonstrate the effect of identically initializing a set of prompts, we conduct experiments on ImageNet-A with 10 initialized prompts in the prompt buffer. The prompts are initialized by the embedding of *“a photo of a”* with random noise. We use our selection strategy to select and optimize the prompts online. When no prompt is selected, we use the initial prompt *“a photo of a”* for prediction.
>
> As shown in the following table, this variant outperforms online TPT and CLIP, which demonstrates that it reduces collapse since it provides diverse prompt options for different test samples through dynamic selection.
> However, it underperforms TPT and our method. The reason can be that the negative influence of previous test samples during online updating is still not entirely solved by the limited number of predefined online prompts, which leads to error accumulation and suboptimal predictions.
> We added this experiment and discussion to Appendix C.
>
>
> | Method     | ImageNet-A  |
> |------------------------|-------|
> | CLIP           | 47.87 |
> | Online TPT          | 6.96 |
> | Identical initialized online prompts  with dynamic selection  | 48.14 |
> | TPT      | 54.77 |
> | ***This paper***    |    **56.17**   |

---

> > ### Comment · Reviewer_kMEy · 2024-11-20
> >
> > I would like to thank the authors for their detailed responses to my questions. I have carefully reviewed their replies, which have addressed all of my concerns. Based on this, I have decided to increase my score.

---

> ### Author Response · Authors · 2024-11-21
>
> Your suggestions help improve our manuscript a lot. Thanks for your updates and prompt encouragement.

---

### Official Review · Reviewer_qkjX · 2024-11-02

**Soundness:** 3
**Presentation:** 3
**Contribution:** 3
**Rating:** 6
**Confidence:** 3

**Summary:**

This paper proposes a new test-time prompt tuning method called DynaPrompt, which leverages a dynamic prompt buffer to extract beneficial information from online test samples while reducing error accumulation. The method adaptively selects and optimizes prompts for each test sample, enabling the selected prompts to integrate relevant information from previous test samples, thus improving the prediction performance of the current sample. Experimental results demonstrate that this method performs effectively across multiple benchmark datasets.

**Strengths:**

1. This paper propose a dynamic prompting method that utilizes useful information from online test samples while mitigating the problem of error accumulation.
2. DynaPrompt improves the adaptive capability of the model by introducing a dynamic prompt selection strategy that adaptively selects and optimizes relevant prompts for each test sample based on two metrics: predictive entropy and probability difference.
3. During the dynamic prompt selection process, if no suitable prompts can be found, DynaPrompt employs a dynamic prompt appending strategy to append new initial prompts to the set of prompts and remove the least active prompts, thus effectively incorporating information from the new data distribution.

**Weaknesses:**

1.  The computational complexity increases, and the authors' approach requires dynamic updating and selection of cues at each stage, whether it introduces more computational time.
2.  The authors' approach demonstrates the advantages of dynamic prompting, however did the authors consider whether comparable performance could also be achieved if online testing was performed from scratch using a prompt length comparable to that of the final model.

**Questions:**

1. It is recommended that the authors add ablation experiments to further demonstrate the validity of the methodology by performing online tests from scratch using the same prompt lengths as the final model and performing performance comparisons.
2. The author's paper mentions deleting the prompt, does the author mean that the entire prompt is deleted in its entirety, or does he mean that only the parameters of the prompt are set to zero?

---

> ### Author Response · Authors · 2024-11-20
> **Response to Reviewer qkjX**
>
> We thank Reviewer qkjX for the constructive feedback and insightful comments. We hope to address the concerns of the reviewer with the responses below.
>
> **Weaknesses**
>
> **Time costs of dynamic selection and appending**
>
> We acknowledge our improved prediction performance comes with an increased computational cost. We note that most of the additional time cost stems from optimizing multiple prompts rather than the selection and appending strategy. For instance, on ImageNet-A, with a prompt buffer size of 10, the total processing time per test sample is approximately 0.39 seconds, of which the selection and appending steps account for only 0.004 seconds. We will clarify this in Section 6.
>
> **Ablations on prompt length**
>
> We thank the reviewer for sharing the insight. To investigate the prompt length effect, we experiment with longer prompts for both TPT and Online TPT. We set the prompt length to 40, which is 10 times longer than the 4-item original *“a photo of a”*. We consider two types of long prompts: (A) 10 times copy of *“a photo of a”*, (B) A prompt generated by GPT-4o: *“Let us solve an image classification task: a photo of a distinct object, animal, plant, or scene, captured in diverse environments and representing meaningful categories. Carefully analyze its features; the exact category of the photo is a”*.
>
> As shown in the following table, longer trainable prompts do not solve the problem of prompt collapse for online learning, even worsening the problem. Since the online testing fails due to error accumulation and prompt collapse, simply improving the length of the prompts does not help. Specifically designed long prompts (B) perform better on the optimization-free CLIP model. However, it may lead to more difficult optimization for test-time tuning, resulting in worse TPT performance. By contrast, based on the prompt selection and appending strategy, our method achieves better performance while reducing error accumulation and prompt collapse. We added the experiments in Appendix C.
>
> | Method         | Initial prompt           | Prompt length | Accuracy |
> |----------------|-----------------------------|-------------------------|----------|
> | CLIP           | *"a photo of a"* 	  | 4                       | 47.87    |
> |                | long prompt (A)             | 40                      | 46.99    |
> |                | long prompt (B)             | 40                      | 48.21    |
> | TPT            | *"a photo of a"*	  | 4                       | 54.77    |
> |                | long prompt (A)             | 40                      | 52.97    |
> |                | long prompt (B)             | 40                      | 52.23    |
> | Online TPT     | *"a photo of a"*	   | 4                       | 6.96     |
> |                | long prompt (A)             | 40                      | 2.06     |
> |                | long prompt (B)             | 40                      | 4.24     |
> | ***This paper*** | *"a photo of a"* 	  | 4 * 10                  | **56.17**    |
>
>
>
>
>
>
> **Questions**
>
> **Prompt deleting**
>
> In our method, deleting means the entire prompt is removed from the buffer, we clarified the text accordingly.

---

> ### Author Response · Authors · 2024-11-25
> **Looking forward to your response**
>
> Dear Reviewer qkjX,
>
> We sincerely thank you for the insightful review. We appreciate the time and effort you put into reviewing our work. We have carefully considered your comments and made improvements based on your suggestions.
> As the discussion period will end in the next two days, please feel free to let us know if you have any further comments. We are willing to engage in further discussion.
>
> Best regards,
>
> Authors

---

> ### Author Response · Authors · 2024-12-02
>
> Dear reviewer qkjX,
>
> I hope this message finds you well. Thank you for your time and efforts in reviewing our submission. Your insights and expertise are greatly appreciated.
>
> We submitted our rebuttal on November 20 and value your evaluation and feedback. As the discussion period is nearing its conclusion in two days, we kindly follow up for your review of our response.
>
> Please feel free to let us know if you have any additional questions to discuss. We are more than willing to provide further clarification or engage in discussion to address any concerns.
>
> Best regards,
>
> Authors

---

### Official Review · Reviewer_FeBK · 2024-11-04

**Soundness:** 3
**Presentation:** 3
**Contribution:** 2
**Rating:** 5
**Confidence:** 5

**Summary:**

This paper addresses fundamental issues in test-time prompt tuning, specifically focusing on the selection, updating, appending, and deletion of prompts. The authors introduce an innovative dynamic test-time prompt tuning approach, which incorporates two novel prompt evaluation metrics alongside a prompt buffer modification strategy. Extensive experimental results underscore the effectiveness of the proposed method.

**Strengths:**

1. The paper is well-structured and easy to follow, with the three technical components clearly and accessibly presented.
2. The proposed method is well-reasoned, employing dynamic prompt selection and updating mechanisms that are both effective and distinct from prior studies, which primarily focus on data manipulation.
3. The experiments are thorough, and the results convincingly demonstrate the effectiveness of the proposed method.

**Weaknesses:**

1. The range of comparison methods could be expanded, as the paper overlooks one relevant comparison method [1].
2. Both the prediction entropy metric and probability difference metric provide insights into model prediction confidence, though from different perspectives. It is unclear why entropy is specifically used to measure relevance while difference is used to maintain diversity. Why does the direct combination of these two types of prompts yield effective results? Would a two-step prompt selection process, satisfying both conditions simultaneously, be more advantageous?
3. Details regarding the data augmentation set  $X_n$  are insufficiently discussed in this paper.

[1] Dingchu Zhang, Zhi Zhou, Yufeng Li: Robust Test-Time Adaptation for Zero-Shot Prompt Tuning. AAAI 2024: 16714-16722

**Questions:**

Please refer to the `Weaknesses` section.

---

> ### Author Response · Authors · 2024-11-20
> **Response to Reviewer FeBK**
>
> We thank Reviewer FeBK for the constructive feedback and insightful comments. We hope to address the concerns of the reviewer with the responses below.
>
> **Weaknesses**
>
> **Comparisons with AdaPrompt**
>
> Thank you for pointing us to this related work on AdaPrompt by Zhang et al. We provide methodological and performance comparisons:
>
> 1) Methodological Comparisons: AdaPrompt performs test-time prompt tuning on a batch of 64 test samples per step, leveraging a buffer to store confident, class-balanced samples for improved tuning and prediction. In contrast, our method dynamically selects and tunes prompts for each single test sample using augmentations.
>
> 2) Performance Comparisons: As the benchmarks in AdaPrompt are not exactly the same as in our paper, we reproduced the method on the missing datasets using their released code. The comparisons are shown in the following tables, with reproduced results indicated in *italics*.
> Our method performs competitively in the cross-dataset setting and outperforms AdaPrompt in the domain generalization setting. This performance gap in the domain generalization setting may arise from the large-scale label space (1000 for ImageNet-V2/S and 200 for ImageNet-A/R), which prevents AdaPrompt's sample buffer from storing class-balanced test samples, leading to degradation. By contrast, our method dynamically tunes the prompt for each sample with its augmentations, therefore achieving consistently good performance.
>
>
> | Method  | Caltech101 | Pets  | Cars  | Flower | Food101 | Aircraft | Sun397 | DTD   | EuroSAT | UCF101 | Mean   |
> |--------------|------------|-------|-------|--------|---------|----------|--------|-------|---------|--------|--------|
> | AdaPrompt | 94.07      | **89.64** | *63.29* | **72.97**  | 84.72   | *21.21*    | *65.37*  | 44.75 | **47.20**    | 67.22  | 65.04 |
> | ***This paper***   | **94.32**      | 88.28 | **67.65** | 69.95  | **85.42**   | **24.33**    | **66.32**  | **47.96** | 42.28   | **68.72**  | **65.52** |
>
> | Method  | Imagenet-v2 | Imagenet-S | Imagenet-A | ImageNet-R | Mean  |
> |--------------|-------------|------------|------------|------------|-------|
> | AdaPrompt | *59.32*       | *47.72*      | *47.71*      | 73.98      | 57.18 |
> | ***This paper***   | **64.67**       | **48.22**      | **56.17**      | **78.17**      | **61.81** |
>
> We highlighted AdaPrompt in Related Work (Section 5) and added the comparisons in Section 6.
>
>
>
> **Dynamic prompt selection metrics and strategy**
>
> Indeed, the proposed two metrics are designed on prediction confidence, but they measure different properties of the model predictions.
> 1) Prediction entropy measures the relevance of the prompt and the sample: A lower prediction entropy means the prompt is more confident in its prediction (Niu et al. 2022; Zhang et al. 2024), indicating the prompt has more prior information about the sample. In this case, the prompt and sample are more relevant to each other.
>
> 2) Probability difference measures the prompt’s sensitivity to the sample augmentations: A larger probability difference means the prompt is more sensitive to the sample augmentations, which implies the prompt predictions are more diverse and effectively avoids over-confident prompts.
>
> We clarify that since we use the measurements of the initial prompt $v_0$ as thresholds to select subsets for both metrics, the selections are independent. By taking the intersection of the two subsets, the selected prompts have both lower entropy and larger probability differences. Therefore, no matter our selection with the two measures, be it separately or sequentially, the results are the same, simultaneously satisfying both conditions. We included these discussions in Section 4.
>
>
> **Details of data augmentation**
>
> We follow the typical data augmentation strategy in TPT (Shu et al. 2022). Specifically, we use AugMix (Hendrycks et al. 2020) to augment the original test image into 63 different augmentation samples, leading to 64 samples in total for each test image. Each test image is first augmented by *“resize”* and random *“crop”*, then fed into the AugMix strategy with several augmentation methods including *auto contrast*, *equalization*, *posterization*, *rotation*, *solarization*, *shearing*, and *translating*. We added the data augmentation clarification in Appendix B.
>
> Hendrycks D, Mu N, Cubuk E D, et al. AugMix: A Simple Data Processing Method to Improve Robustness and Uncertainty. ICLR 2020.

---

> ### Author Response · Authors · 2024-11-25
> **Looking forward to your response**
>
> Dear Reviewer FeBK,
>
> We sincerely thank you for the insightful review. We appreciate the time and effort you put into reviewing our work. We have carefully considered your comments and made improvements based on your suggestions.
> As the discussion period will end in the next two days, please feel free to let us know if you have any further comments. We are willing to engage in further discussion.
>
> Best regards,
>
> Authors

---

> ### Author Response · Authors · 2024-12-02
>
> Dear reviewer FeBK,
>
> I hope this message finds you well. Thank you for your time and efforts in reviewing our submission. Your insights and expertise are greatly appreciated.
>
> We submitted our rebuttal on November 20 and value your evaluation and feedback. As the discussion period is nearing its conclusion in two days, we kindly follow up for your review of our response.
>
> Please feel free to let us know if you have any additional questions to discuss. We are more than willing to provide further clarification or engage in discussion to address any concerns.
>
> Best regards,
>
> Authors

---

### Official Review · Reviewer_tKkt · 2024-11-04

**Soundness:** 3
**Presentation:** 3
**Contribution:** 3
**Rating:** 6
**Confidence:** 4

**Summary:**

This paper introduces a dynamic test-time prompt tuning approach that enhances zero-shot generalization in vision-language models. It leverages the beneficial information from previous online samples to adaptively select and optimize prompts, reducing error accumulation and improving model performance across various datasets.

**Strengths:**

1. The paper is well-organized and easy to follow.
2. The proposed DynaPrompt effectively mitigates error accumulation, a prevalent challenge in online test-time tuning，leading to more stable performance across sequential test samples while exploiting beneficial information from prior online test samples.
3. Despite the increased time costs associated with larger prompt buffer sizes, the experimental outcomes confirm the  effectiveness of the proposed method.

**Weaknesses:**

1. In prompt learning, the initial prompts might affect the final performance. I wonder whether a similar situation can occur with the proposed method. The authors are encouraged to conduct related experiments.
2. Could the proposed method be extended to incorporate visual prompts, thereby evolving into a multimodal approach? Additionally, when integrated with MaPLe, is the method only applied to the textual branch?

**Questions:**

1. I am curious about how the order for each round in the 'Sensitivity to test time sample order' section was set.

---

> ### Author Response · Authors · 2024-11-20
> **Response to Reviewer tKkt**
>
> We thank Reviewer tKkt for the constructive feedback and insightful comments. We hope to address the concerns of the reviewer with the responses below.
>
> **Weaknesses**
>
> **Effect of initial prompts**
>
> Thank you for sharing the insight. We conducted experiments on ImageNet-A using various initial text prompts. As shown in the following table, the initial prompts affect the performance of CLIP (Radford et al. 2021), TPT (Shu et al. 2022), as well as our method. The reason can be related to the initial predictions of the original CLIP model. Nonetheless, our method consistently outperforms TPT, showing robustness despite variations in initialization. We added this experiment and discussions in Appendix C.
>
> | Initial prompt          | CLIP  | TPT   | *This paper* |
> |------------------------|-------|-------|------------|
> | a photo of a           | 47.87 | 54.77 | **56.17**      |
> | an image of a          | 48.31 | 54.84 | **56.19**      |
> | high-quality of a      | 44.48 | 51.47 | **52.57**      |
> | Identify feature of    | 46.08 | 50.18 | **51.84**      |
> | Visual features of the | 46.21 | 51.48 | **53.13**      |
> | natural photo of a     | 44.82 | 52.56 | **54.08**      |
> | classify the photo of  | 46.60  | 52.51 | **53.27**      |
> | average                | 46.35 | 52.54 | **53.89**      |
>
>
> **Multimodal test-time prompt tuning**
>
> We clarify that our approach can be evolved into a multimodal setting based on MaPLe. When integrated with MaPLe, our dynamic test-time prompt tuning is already applied on *both* the textual and visual branches. We clarified the corresponding implementation details in Section 6.
>
> **Questions**
>
> **How to set sample orders during dynamic prompt tuning**
>
> We shuffle the sample order in the dataloader with different random seeds at test time, which leads to different sample orders. We added this description in Section 6 and will release the corresponding PyTorch code.

---

> ### Comment · Reviewer_tKkt · 2024-11-24
>
> Thanks for your responses. Most of my concerns have been addressed and I would like to maintain my original rating. This is a meaningful work, which tends to be accepted.

---

> ### Author Response · Authors · 2024-11-25
>
> Thanks for your updates and encouragement. Your suggestions have helped us improve the manuscript.

---

### Meta-Review · Area_Chair_5sPi · 2024-12-18

**Metareview:**

The paper tackles the problem of test-time prompt tuning with a new approach that adaptively updates samples in the test-time learning pool and thus improves sample quality for test-time prompt tuning. The paper received four reviews of 1x accept, 2x borderline accept, and 1x borderline reject ratings. In general, the reviews are positive. The reviewers appreciated the idea of dynamic test-time prompt tuning and acknowledged the effectiveness as evidenced by the results. In the meantime, the reviewers had questions about the increased computational cost caused by the dynamic learning mechanism and requested more results using different backbones and more baselines. The authors had properly addressed these during the rebuttal. The proposed method is novel and the findings are valuable to the community. Therefore, the AC recommends that the paper be accepted.

**Additional Comments On Reviewer Discussion:**

In the first-round review, the reviewers requested that the authors expand the comparisons to include more methods and analysis, as well as more results using different backbones. The rebuttal provided additional results to support the paper. Reviewer kMEy engaged in the rebuttal discussion and increased the score to accept. Reviewer FeBK did not engage. The AC has checked the rebuttal and found that the rebuttal has done good job in addressing these concerns.

---

### Decision · Program_Chairs · 2025-01-22

Accept (Poster)